# Properties of winning Iterated Prisoner's Dilemma strategies

**Nikoleta E. Glynatsi**[1,2]*, **Vincent Knight**[2], **Marc Harper**[3]

**1** Research Group Dynamics of Social Behavior, Max Planck Institute for Evolutionary Biology, Plön, Germany, **2** School of Mathematics, Cardiff University, Cardiff, United Kingdom, **3** Google Inc., Mountain View, California, United States of America

☯ These authors contributed equally to this work.
\* glynatsi@evolbio.mpg.de

**Data Availability Statement:** The raw and processed datasets are:publicly available on Zenodo at https://zenodo.org/records/10246248 and https://zenodo.org/records/10246247. These datasets can be used for further analysis and insights. The code used to perform the analysis is

## Abstract

Researchers have explored the performance of Iterated Prisoner's Dilemma strategies for decades, from the celebrated performance of Tit for Tat to the introduction of the zero-determinant strategies and the use of sophisticated learning structures such as neural networks. Many new strategies have been introduced and tested in a variety of tournaments and population dynamics. Typical results in the literature, however, rely on performance against a small number of somewhat arbitrarily selected strategies, casting doubt on the generalizability of conclusions. In this work, we analyze a large collection of 195 strategies in thousands of computer tournaments, present the top performing strategies across multiple tournament types, and distill their salient features. The results show that there is not yet a single strategy that performs well in diverse Iterated Prisoner's Dilemma scenarios, nevertheless there are several properties that heavily influence the best performing strategies. This refines the properties described by Axelrod in light of recent and more diverse opponent populations to: be nice, be provocable and generous, be a little envious, be clever, and adapt to the environment. More precisely, we find that strategies perform best when their probability of cooperation matches the total tournament population's aggregate cooperation probabilities. The features of high performing strategies help cast some light on why strategies such as Tit For Tat performed historically well in tournaments and why zero-determinant strategies typically do not fare well in tournament settings.

## Author summary

In 1980, political scientist Robert Axelrod ran one of the most famous computer tournaments of the Iterated Prisoner's Dilemma (IPD). The winner? The now-famous strategy, Tit for Tat. Axelrod attributed its success to simple properties such as: do not be envious, avoid being the first to defect, and do not be overly clever. Yet the tournament design, using only a small, selected set of strategies, not including random noise, and having fixed game lengths, raises questions about the generalizability of these results. Many researchers have continued to make similar assumptions in their own IPD experiments, limiting the insights that can be applied to more complex, realistic settings.

available in the online GitHub repository: Nikoleta-v3/meta-analysis-of-prisoners-dilemma-tournaments.

**Funding:** N.E.G. acknowledges the generous support of the European Research Council Starting Grant 850529: E-DIRECT and the Max Planck Society. The funders had no role in study design, data collection and analysis, decision to publish, or preparation of the manuscript.

**Competing interests:** The authors have declared that no competing interests exist.

In our study, we address these limitations by analyzing the performance of a large and diverse collection of IPD strategies across thousands of computer tournaments. We find that, while no single strategy consistently excels, successful strategies share key characteristics: they are nice, provocable and generous, a little envious, clever, and adapt to the environment. More precisely, strategies perform best when their probability of cooperation matches the total tournament population's aggregate cooperation probabilities.

## Introduction

The Iterated Prisoner's Dilemma (`IPD`) is a repeated two-player game that models behavioral interactions, specifically interactions where self-interest clashes with collective interest. It encompasses a wide range of social and biological phenomena. In each turn of the game, both players simultaneously and independently decide between cooperation ($C$) and defection ($D$). This decision is made with the memory of all prior interactions. The payoffs for each player at each turn are influenced by their own choice and the choice of the other player. To this end, the payoffs of the game are defined by

$$\begin{array}{c} \\ \text{Cooperate } (C) \\ \text{Defect } (D) \end{array} \begin{array}{cc} \text{Cooperate } (C) & \text{Defect } (D) \\ \begin{pmatrix} R, R & S, T \\ T, S & P, P \end{pmatrix} \end{array}. \tag{1}$$

where typically $T > R > P > S$ and $2R > T + S$. The most common values used in the literature [1] are $R = 3$, $P = 1$, $T = 5$, $S = 0$, and these are the values also used in this work.

Conceptualizing strategies and understanding the best way to play the game have been of interest to the scientific community since the formulation of the game [2–13]. This extends to both tournament settings and population dynamics. Computer tournaments became a common evaluation technique for newly designed strategies following Axelrod's computer tournaments in the 1980s [2, 14]. The winner of both of Axelrod's tournaments [2, 14] was the simple strategy Tit For Tat (`TFT`). `TFT` cooperates on the first turn and thereafter copies the previous action of its opponent, retaliating against defections with a defection and forgiving a defection if followed by cooperation. Axelrod concluded that the strategy's robustness was due to four properties, which he adapted into four suggestions for success in an `IPD` tournament:

1. Do not be envious by striving for a payoff larger than the opponent's payoff.

2. Be "nice" by not being the first to defect.

3. Reciprocate both cooperation and defection; Be provocable to retaliation and forgiveness.

4. Do not be too clever by scheming to exploit the opponent.

Forgiveness, in this context, is a strategy's ability to cooperate after a $DC$ outcome to achieve mutual cooperation again. In environments without noise, `TFT` would end up in $DC$ only if it had received a defection and then retaliated. Subsequently, `TFT` would forgive an opponent that apologizes (in a $DC$ round) by returning to cooperation, as mutual cooperation is deemed better than mutual defection.

Due to the strategy's strong performance in both tournaments and a series of evolutionary experiments [1], `TFT` was often claimed to be a highly robust (and sometimes the most robust) strategy for the `IPD`. There are strategies that have built upon `TFT` and the reciprocity-based approach. In [5], a strategy called Gradual was introduced, constructed to have the same

qualities as those of `TFT` with one addition. Gradual has a memory of the previous rounds of play in the game, recording the number of defections by the opponent and punishing them with a growing number of defections. It then enters a calming state in which it cooperates for two rounds. A strategy with the same intuition as Gradual is Adaptive Tit for Tat [15]. Adaptive Tit for Tat maintains a continually updated estimate of the opponent's behavior and uses this estimate to condition its future actions. Other research has built upon the limitations of `TFT`. For example, in [16–19], it was shown that `TFT` suffered in environments with noise. This was mainly due to the strategy being too provocable and its lack of generosity and contrition. Since `TFT` immediately punishes a defection, in a noisy environment, it can get stuck in a repeated cycle of defections and cooperations. Some new strategies, more robust in tournaments with noise, were soon introduced, including Nice and Forgiving [16], Generous Tit For Tat [3], and Pavlov (aka Win Stay Lose Shift) [4], as well as later variants such as OmegaTFT [20].

Finally, others introduced strategies deviating completely from the originally suggested properties of success. For example, a set of "envious" Iterated Prisoner's Dilemma (`IPD`) strategies were introduced, called zero-determinant strategies (ZDs), in [6]. These strategies attempt to force a linear relationship between stationary payoffs against their opponents, potentially ensuring that they receive a higher average payout. While ZDs were introduced with a small tournament in which some were reportedly successful [21], this result has not generally held in future work [22]. Furthermore, in [23], a series of "clever" strategies trained using reinforcement learning were introduced. These strategies were trained using lookup tables [24], hidden Markov models [23], and finite-state automata [25], on a set of 170 strategies.

One thing that has remained the same is that the introduction of a new strategy is often accompanied by a claim that the new strategy is the best performing strategy for the `IPD`, often without extensive testing against a broad spectrum of opponents or representative classes of opponents. The lack of testing against formally defined strategies and tournament winners is understandable given the effort required to implement the hundreds of published `IPD` strategies. Implementing prior strategies faithfully is often extremely difficult or impossible due to insufficient descriptions and lack of published implementations or code. Despite these challenges, the absence of thorough testing raises concerns about claims regarding the superiority or robustness of newly introduced strategies.

Beyond these difficulties, we believe that limited comprehensive analyses are rooted in field conventions. Tournaments or evolutionary dynamics often rely on a select list of hand-picked strategies chosen by modelers, typically based on specific properties they wish to examine. This practice may stem from misconceptions, such as the assumption that because `TFT` performs relatively well, it is sufficient to test only against `TFT` variants. Another misconception may stem from the Press & Dyson result [6], which implies that lower memory strategies will always dominate pairwise interactions, leading some to consider only memory-one strategies. However, this result holds strictly only in pairwise interactions.

It is not only the set of strategies or the tournament parameters such as noise that may impact results but also the design of the round-robin tournament itself. To address this [26] separately examined the effects of changes in format, objective criteria, and payoff values on tournament outcomes. They demonstrated that `TFT`'s performance declined under certain conditions. To our knowledge, this is the only study that has reanalyzed the tournament structure and critically evaluated it. However, in this work, we employ an extensive list of strategies made possible by the `Axelrod-Python` package, an approach that would have been difficult to achieve previously. Unlike the authors of that study, we do not consider a new tournament design.

In this paper, we evaluate the performance of a significant number of `IPD` strategies across a diverse array of tournaments. Many of the strategies used in our analysis are drawn from well-known and named strategies in `IPD` literature, including previous tournament winners. This contrasts with other work that is often constrained to specific classes such as as memory-one strategies or those of a certain structural form like finite state machines or deterministic memory-two strategies. Furthermore, our tournaments encompass variations, including standard tournaments resembling Axelrod's original ones, tournaments with noise, probabilistic match length, and both noise and probabilistic match length. This diversity in strategies and tournament types provides new insights and tests earlier claims in alternative settings against known powerful strategies. More specifically, we show that the previous tournament winners are lacking against large enough opponent pools; they do not appear among the top-performing strategies anymore. This could be due to likely suffering from a lack of diversity in the strategies they were trained/tested against, finding it hard to adapt to the new strategies.

It is important to note that we do not assert the existence of a single best-performing strategy across all tournaments or tournament types. On the contrary, our work demonstrates that such a strategy does not exist (notwithstanding a few strategies with broadly high performance). The primary objective of this paper, presented in the latter parts of the paper, is to continue the discussion on the properties of successful strategies, a conversation started by Axelrod. The results of our analysis conclude that the properties of a successful strategy in the Iterated Prisoner's Dilemma (`IPD`) are:

1. Be a little bit envious

2. Be "nice" in non-noisy environments or when game lengths are longer

3. Reciprocate both cooperation and defection appropriately; Be provocable in tournaments with short matches, and generous in tournaments with noise

4. It's ok to be clever

5. Adapt to the environment; Adjust to the mean population cooperation

We believe that the discussion on the properties of winning strategies holds significant importance. It aims to provide guidance to researchers designing new strategies and those training strategies. Specifically, much like the recognized value of diversity in training datasets [27], such as variations in image perspective, skin color, etc., are critical in training accurate and generalizable machine learning models, we show that diversity in the population of opponent strategies is of paramount importance in the construction and evaluation of game theory strategies. Moreover, conducting a similar analysis can shed light on already trained strategies, aiding in understanding the key features they have autonomously developed during their training processes.

## Model

The data collection of various types of tournaments and the use of different strategies are made possible due to an open-source library called `Axelrod-Python` [28] (version 3.0.0). `Axelrod-Python` enables the simulation of `IPD` tournaments and contains an extensive list of strategies. Most of these strategies are described in the literature, with a few exceptions contributed specifically to the package. In this paper, we use a total of 195 strategies, which can be found in the Supplementary Material (S1 Text). The package supports several tournament types, and this work considers standard, noisy, probabilistic ending, and noisy probabilistic ending tournaments.

```
>>> import axelrod as axl

>>> players = [axl.TitForTat(), axl.GTFT(), axl.Gradual(), axl.EvolvedLookerUp2_2_2()]

>>> standard_tournament = axl.Tournament(players=players, turns=50, repetitions=10, noise=0)

>>> results = standard_tournament.play()

>>> prob_noisy_end_tournament = axl.Tournament(players=players,
...                                            prob_end=0.5,
...                                            repetitions=10,
...                                            noise=0.001)

>>> results = prob_noisy_end_tournament.play()
```

**Fig 1. Example usage of the `Axelrod-Python` package for running tournaments.** The strategies that players use are saved in a list, which is then passed to the Tournament class. In this example, strategies we previously discussed, such as `TFT`, Generous Tit-for-Tat, Gradual, and a stochastic strategy evolved through reinforcement learning, are included. We create a standard tournament where `noise` is set to 0, as well as a noisy tournament with probabilistic ending. Once an instance of the tournament class is defined, executing the tournament is straightforward. The `play()` method generates all possible pairs from the list of strategies, including pairs where each strategy plays against itself, and then iterates through each match to play it. Each match is repeated according to the specified number of repetitions, with results aggregated to a tournament summary.

*Standard tournaments* are similar to Axelrod's well-known tournaments [2]. In these tournaments, there are $N$ strategies, and each strategy plays an iterated game with $n$ turns against all other strategies, not including self-interactions. *Noisy tournaments* also involve $N$ strategies and $n$ turns, but in each turn, there is a probability $p_n$ that a player's action is flipped. Compared to these two tournaments, in *probabilistic ending tournaments* the number of turns is not fixed. Instead, a match between strategies ends with a given probability $p_e$. Finally, *noisy probabilistic ending tournaments* incorporate both a noise probability $p_n$ and an ending probability $p_e$. For smoother results, each tournament is repeated $k$ times, and this repetition factor was allowed to vary to assess the impact of smoothing. The winner of each tournament is determined based on the average score achieved by a strategy from the entire set of repetitions, not by the number of wins.

To run a tournament, only a few lines of code are required (Fig 1). Specifically, one needs to define the list of strategies that the players use when participating in the tournament, the number of repetitions, the number of turns or the probability of each match ending, and the probability of noise. We demonstrate two examples in Fig 1: one with a fixed number of turns and one with a probabilistic ending. A tournament is an instance of the `Tournament` class. To execute the tournament, users need to run the `play()` method, which returns an instance of the `ResultSet` class. This instance includes many details of the tournament, such as the winner and the average score of the participants. Additionally, the instance contains a more detailed summary of the tournament, which we use in our analysis. We describe the results summary in detail below.

The process of collecting tournament results is outlined in Algorithm 1. For each trial, a random size $N$ is selected, and a random list of $N$ strategies from the 195 available. Subsequently, one standard, one noisy, one probabilistic ending, and one noisy probabilistic ending tournament are conducted for the selected list of strategies. The parameters for the tournaments, as well as the number of repetitions, are chosen once for each trial. We have run a total of 11400 trials of Algorithm 1. For each trial, we collect the results for four different tournaments, resulting in a total of 45600 (11400 × 4) tournament results. Each tournament outputs a result summary in the form of Table 1.

**Table 1. Result summary example of a tournament.** A result summary consists of *N* rows, with each row containing information for each strategy that participated in the tournament. This information includes the strategy's rank (*R*), median score, the cooperation rate (*C_r*), the number of match wins, and the probability that the strategy cooperated in the opening move. Additionally, it provides the probabilities of a strategy being in any of the four states (*CC, CD, DC, DD*) and the cooperation rate after each state.

| Rank | Name | Median score | Cooperation rating ($C_r$) | Win | Initial C | Rates | | | | | | | |
|------|------|--------------|------------------------------|-----|-----------|-------|------|------|------|---------|---------|---------|---------|
| | | | | | | CC | CD | DC | DD | CC to C | CD to C | DC to C | DD to C |
| 0 | EvolvedLookerUp2 2 2 | 2.97 | 0.705 | 28.0 | 1.0 | 0.639 | 0.066 | 0.189 | 0.106 | 0.836 | 0.481 | 0.568 | 0.8 |
| 1 | Evolved FSM 16 Noise 05 | 2.875 | 0.697 | 21.0 | 1.0 | 0.676 | 0.020 | 0.135 | 0.168 | 0.985 | 0.571 | 0.392 | 0.07 |
| 2 | PSO Gambler 1 1 1 | 2.874 | 0.684 | 23.0 | 1.0 | 0.651 | 0.034 | 0.152 | 0.164 | 1.000 | 0.283 | 0.000 | 0.136 |
| 3 | PSO Gambler Mem1 | 2.861 | 0.706 | 23.0 | 1.0 | 0.663 | 0.042 | 0.145 | 0.150 | 1.000 | 0.510 | 0.000 | 0.122 |
| 4 | Winner12 | 2.835 | 0.682 | 20.0 | 1.0 | 0.651 | 0.031 | 0.141 | 0.177 | 1.000 | 0.441 | 0.000 | 0.462 |
| . . . | . . . | . . . | . . . | . . . | . . . | . . . | . . . | . . . | . . . | . . . | . . . | . . . | . . . |

**Algorithm 1**: Tournament Data Collection Algorithm

```
for seed ∈ [0, 11420] do
    N ← randomly select integer ∈ [3, 195];
    players ← randomly select N players;
    k ← randomly select integer ∈ [10, 100];
    n ← randomly select integer ∈ [1, 200];
    pₙ ← randomly select float ∈ [0, 1];
    pₑ ← randomly select float ∈ [0, 1];
    result standard ← Axelrod.tournament(players, n, k);
    result noisy ← Axelrod.tournament(players, n, pₙ, k);
    result probabilistic ending ← Axelrod.tournament(players, pₑ, k);
    result noisy probabilistic ending ← Axelrod.tournament(players, pₙ,
pₑ, k);
return result standard, result noisy, result probabilistic ending,
result noisy probabilistic ending;
```

The summary contains statistics regarding each strategy that participated in the tournament, such as its rank, cooperation rate, time spent in each state when only a single past round is considered, and the probability of cooperating after each of the four possible outcomes of the previous round. In our analysis, we will use the measures for each strategy provided in the summary, as well as additional measures we calculated. Namely, these include the SSE error, the average, median, maximum, and minimum cooperation rates in each tournament. The SSE (introduced in [29]) shows how closely a strategy behaves as a zero-determinant strategy and subsequently in an extortionate way. We also consider how each strategy's cooperation rate $C_r$ compares to those of the tournament as a whole, for example, by comparing $C_r$ to $C_{max}$.

During the data collection process, the probabilities of noise ($p_n$) and tournament ending ($p_e$) were allowed to take values between 0 and 1. However, commonly used values for these probabilities are $p_n \leq 0.1$ and $p_e \leq 0.1$. This is to make the results more interpretable. For example, consider a strategy competing in an environment with $p_n > 0.1$. In cases with a high value of noise, most of the actions the strategy takes are the complete opposite of what the strategy is designed to do. Therefore, we will focus on the tournaments for which $p_n \leq 0.1$ and $p_e \leq 0.1$. Thus, the results presented here pertain to subsets of the noisy and probabilistic ending tournaments. Specifically, the results rely on 1150 tournaments with noise, 1134 tournaments with a probabilistic ending, and 117 tournaments with both noise and a probabilistic ending. We also provide an analysis of the paper considering the entire datasets, and these results are presented in the Supplementary Material (S1 Text). The general results of the analysis are not affected by the restriction of the noise and probabilistic ending probabilities.

## Results

### Top ranked strategies across tournaments

A strategy has participated in multiple tournaments of each type, and to evaluate its overall performance, we introduce a measure called the *normalized rank*. In each tournament, the strategies receive a rank ($R$), where 0 denotes that the strategy was the winner, and $N-1$ indicates that the strategy came last in the tournament. The normalized rank, denoted as $r$, is calculated as $r = \frac{R}{N-1}$. Thus, the rank a strategy achieved over the number of players in the tournament. The performance of the strategies is assessed based on the *median of the normalized rank*, denoted as $\bar{r}$.

For example, let's consider the well-known strategies TFT and Gradual. Each strategy participated in several tournaments of each type. In Fig 2 we show the distribution of the normalised ranks of these strategies in each of the four tournaments. We can observe that TFT looks to be normally distributed normalized rank. In comparison, Gradual's performance has longer tails, indicating that there were tournaments where the strategy performed very well or very poorly. Overall, Gradual achieves a lower median rank, signifying that it performs better than TFT except in the case of noisy and probabilistic ending tournaments (lower rank is better).

The top 15 strategies for each tournament type, based on $\bar{r}$, are presented in Table 2, while the $r$ distributions for the top-ranked strategies can be found in Fig 3.

In standard tournaments dominating strategies were those trained using reinforcement learning techniques. 10 out of the 15 top strategies were introduced in [23]. These strategies are based on finite state automata (FSM), hidden Markov models (HMM), artificial neural networks (ANN), lookup tables (LookerUp), and stochastic lookup tables (Gambler). They have been trained using reinforcement learning algorithms (evolutionary and particle swarm algorithms) to perform well against a subset of the strategies in Axelrod-Python in a standard

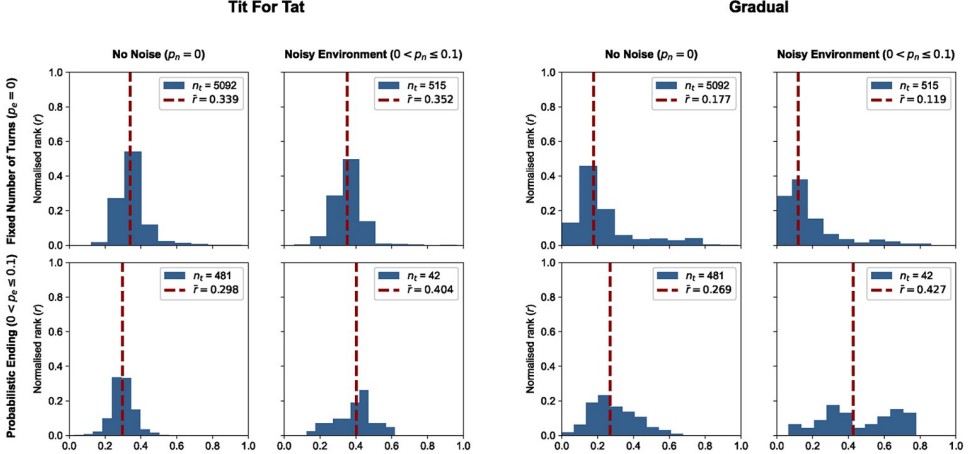

**Fig 2. Examples of normalized rank distributions for two strategies, TFT and Gradual.** We plot the distributions of *r* for the two strategies in the four tournament types. As a reminder, lower values of *r* correspond to better performances. The top left quadrant of each plot shows the distribution for standard tournaments (fixed number of turns and no noise). The top right quadrant shows the distribution for noisy tournaments (fixed number of turns and noise). The bottom left quadrant shows the distribution for probabilistic ending tournaments (no noise and probabilistic ending). Finally, the bottom right quadrant shows the distribution for noisy probabilistic ending tournaments (noise and probabilistic ending). In each quadrant, we also show the number of data points. Both strategies participated in a similar number of tournaments. Based on the median rank, which we use in this work to define overall performance, TFT performs best in probabilistic ending tournaments, whereas Gradual was in standard tournaments.

**Table 2. Top performances for each tournament type based on $\bar{r}$.** The results of each type are based on 11420 unique tournaments. The results for noisy tournaments with $p_n < 0.1$ are based on 1151 tournaments, and for probabilistic ending tournaments with $p_e < 0.1$ on 1139. The top ranks indicate that trained strategies perform well in a variety of environments, but so do simple deterministic strategies. For noisy tournaments DBS is the top ranked strategy with $\bar{r} = 0$, thus DBS won every tournament it participated in. The same for Evolved FSM 16 Noise 05 in probabilistic ending.

| | Standard | | Noisy ($p_n \leq 0.1$) | | Probabilistic ending ($p_e < 0.1$) | | Noisy probabilistic ending | |
|---|---|---|---|---|---|---|---|---|
| | Name | $\bar{r}$ | Name | $\bar{r}$ | Name | $\bar{r}$ | Name | $\bar{r}$ |
| 0 | Evolved HMM 5 | 0.007 | DBS | 0.0 | Evolved FSM 16 | 0.0 | Raider | 0.022 |
| 1 | Evolved FSM 16 | 0.01 | Evolved FSM 16 Noise 05 | 0.008 | Evolved FSM 16 Noise 05 | 0.013 | MEM2 | 0.037 |
| 2 | EvolvedLookerUp2_2_2 | 0.011 | Evolved ANN 5 Noise 05 | 0.013 | MEM2 | 0.027 | Prober 3 | 0.039 |
| 3 | Evolved FSM 16 Noise 05 | 0.017 | BackStabber | 0.024 | Evolved HMM 5 | 0.043 | Evolved FSM 16 Noise 05 | 0.048 |
| 4 | PSO Gambler 2_2_2 | 0.022 | DoubleCrosser | 0.025 | EvolvedLookerUp2_2_2 | 0.049 | Hard Prober | 0.072 |
| 5 | Evolved ANN | 0.029 | Evolved ANN 5 | 0.028 | Spiteful Tit For Tat | 0.059 | Spiteful Tit For Tat | 0.078 |
| 6 | Evolved ANN 5 | 0.034 | Evolved ANN | 0.038 | Nice Meta Winner | 0.069 | Better and Better | 0.089 |
| 7 | PSO Gambler 1_1_1 | 0.037 | Spiteful Tit For Tat | 0.051 | NMWE Finite Memory | 0.069 | Grudger | 0.091 |
| 8 | Evolved FSM 4 | 0.049 | Evolved HMM 5 | 0.051 | NMWE Deterministic | 0.07 | Fortress4 | 0.096 |
| 9 | PSO Gambler Mem1 | 0.05 | Level Punisher | 0.052 | Grudger | 0.07 | Meta Winner Memory One | 0.099 |
| 10 | Winner12 | 0.06 | Omega TFT | 0.059 | NMWE Long Memory | 0.074 | NMWE Long Memory | 0.099 |
| 11 | Fool Me Once | 0.061 | Fool Me Once | 0.059 | Nice Meta Winner Ensemble | 0.076 | Nice Meta Winner | 0.104 |
| 12 | DBS | 0.071 | PSO Gambler 2_2_2 Noise 05 | 0.067 | EvolvedLookerUp1_1_1 | 0.077 | NMWE Deterministic | 0.109 |
| 13 | DoubleCrosser | 0.072 | Evolved FSM 16 | 0.078 | NMWE Memory One | 0.08 | NMWE Memory One | 0.112 |
| 14 | BackStabber | 0.075 | EugineNier | 0.08 | NMWE Stochastic | 0.085 | Nice Meta Winner Ensemble | 0.115 |

tournament. Thus, their performance in the specific setting was anticipated, although still noteworthy given the random sampling of tournament participants. DoubleCrosser and Back-Stabber, both from the `Axelrod-Python`, use the number of turns and are set to defect in the last two rounds. These strategies can be characterized as "cheaters" because their source code allows them to know the number of turns (unless the match has a probabilistic ending). These strategies were expected to not perform as well in tournaments where the number of turns is not specified. Finally, Winner 12 [22] and DBS [30] are both from the literature. DBS is a strategy specifically designed for noisy environments; however, it ranks highly in standard tournaments as well. Similarly, the fourth-ranked player, Evolved FSM 16 Noise 05, was trained for noisy tournaments yet performs well in standard tournaments.

In the case of noisy tournaments, the top-performing strategies include strategies specifically designed for noisy tournaments. These are DBS, Evolved FSM 16 Noise 05, Evolved ANN 5 Noise 05, PSO Gambler 2 2 2 Noise 05, and Omega Tit For Tat [20]. Omega TFT, a strategy designed to break the deadlocking cycles of *CD* and *DC* that TFT can fall into in noisy environments, places 10th. The rest of the top ranks are occupied by strategies that performed well in standard tournaments and deterministic strategies such as Spiteful Tit For Tat [31], Level Punisher [32], Eugine Nier [33].

Furthermore, in tournaments with probabilistic endings, the highly ranked strategies leaned towards defecting strategies and trained finite state automata, as demonstrated by the works of Ashlock et al. [34, 35]. The most effective strategies in probabilistic ending tournaments are also a series of ensemble Meta strategies, trained strategies that performed well in standard tournaments, and Grudger [28] and Spiteful Tit for Tat [31]. The Meta strategies [28] utilize a team of strategies and aggregate the potential actions of the team members into a single action in various ways.

While no single strategy consistently outperforms all others in any of the distinct tournament types or across various tournament types, certain types of strategies consistently achieve

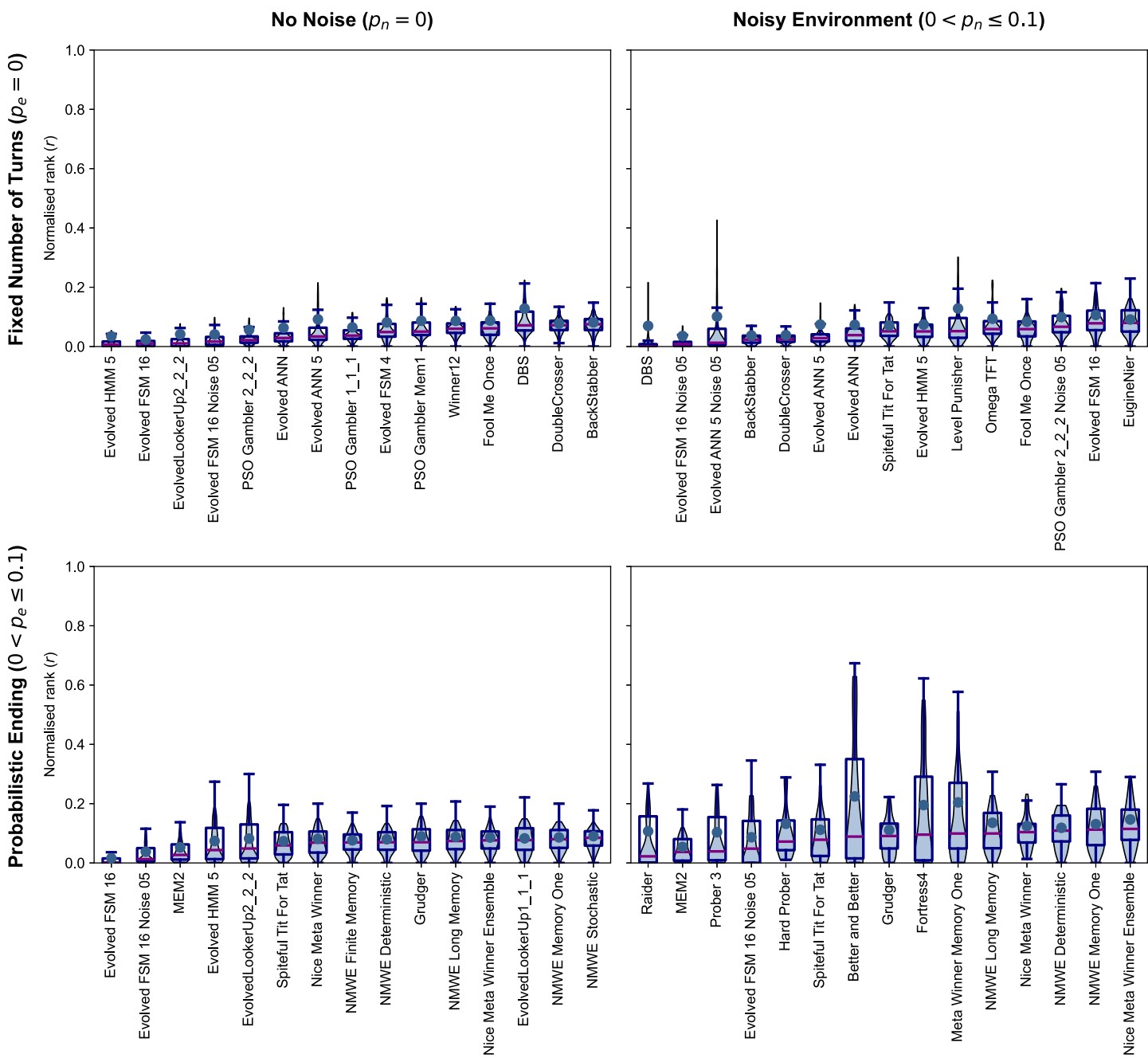

**Fig 3. $r$ distributions of the top 15 strategies in different environments.** A lower value of $\bar{r}$ corresponds to a more successful performance. A strategy's $r$ distribution skewed towards zero indicates that the strategy ranked highly in most tournaments it participated in. Most distributions are skewed towards zero.

top rankings. These include strategies that have undergone training, those that retaliate, and those that adapt their behavior based on preassigned rules to optimize outcomes. These findings challenge some of Axelrod's suggestions, particularly the advice to "not be clever" and "not be envious".

**Table 3. Included features for performance evaluation analysis.** Stochastic, makes use of length and makes use of game are APL classifiers that determine whether a strategy is stochastic or deterministic, whether it makes use of the number of turns or the game's payoffs. The memory usage is calculated as the number of turns the strategy considers to make an action (which is specified in the APL) divided by the number of turns. The SSE (introduced in [29]) shows how close a strategy is to behaving as a ZDs, and subsequently, in an extortionate way. The method identifies the ZDs closest to a given strategy and calculates the algebraic distance between them as the sum of squared error (SSE). A SSE value of 1 indicates no extortionate behaviour at all whereas a value of 0 indicates that a strategy is behaving as a ZDs. The memory usage of strategies is the number of rounds of play used by the strategy when deciding on an action, divided by the number of turns in each match. For example, Winner12 uses the previous two rounds of play, and if participating in a match with 100 turns its memory usage would be 2/100. For strategies with an infinite memory size, for example Evolved FSM 16 Noise 05, memory usage is equal to 1. Note that for tournaments with a probabilistic ending the number of turns was not collected, so the memory usage feature is not used for probabilistic ending tournaments. The rest of the features considered are the $CC$ to $C$, $CD$ to $C$, $DC$ to $C$, and $DD$ to $C$ rates as well as cooperating ratio of a strategy, the minimum ($C_{min}$), maximum ($C_{max}$), mean ($C_{mean}$) and median ($C_{median}$) cooperating ratios of each tournament.

| feature | feature explanation | source | value type | min value | max value |
|---|---|---|---|---|---|
| stochastic | If a strategy is stochastic | strategy classifier from APL | boolean | Na | Na |
| makes use of game | If a strategy makes used of the game information | strategy classifier from APL | boolean | Na | Na |
| makes use of length | If a strategy makes used of the number of turns | strategy classifier from APL | boolean | Na | Na |
| memory usage | The memory size of a strategy divided by the number of turns | memory size from APL | float | 0 | 1 |
| SSE | A measure of how far a strategy is from ZD behaviour | method described in [29] | float | 0 | 1 |
| max cooperating rate ($C_{max}$) | The biggest cooperating rate in a given tournament | result summary | float | 0 | 1 |
| min cooperating rate ($C_{min}$) | The smallest cooperating rate in a given tournament | result summary | float | 0 | 1 |
| median cooperating rate ($C_{median}$) | The median cooperating rate in a given tournament | result summary | float | 0 | 1 |
| mean cooperating rate ($C_{mean}$) | The mean cooperating rate in a given tournament | result summary | float | 0 | 1 |
| $C_r / C_{max}$ | A strategy's cooperating rate divided by the maximum | result summary | float | 0 | 1 |
| $C_{min} / C_r$ | The minimum divided by a strategy's cooperating rate | result summary | float | 0 | 1 |
| $C_r / C_{median}$ | A strategy's cooperating rate divided by the median | result summary | float | 0 | 1 |
| $C_r / C_{mean}$ | A strategy's cooperating rate divided by the mean | result summary | float | 0 | 1 |
| $C_r$ | The cooperating ratio of a strategy | result summary | float | 0 | 1 |
| $CC$ to $C$ rate | The probability a strategy will cooperate after a mutual cooperation | result summary | float | 0 | 1 |
| $CD$ to $C$ rate | The probability a strategy will cooperate after being betrayed by the opponent | result summary | float | 0 | 1 |
| $DC$ to $C$ rate | The probability a strategy will cooperate after betraying the opponent | result summary | float | 0 | 1 |
| $DD$ to $C$ rate | The probability a strategy will cooperate after a mutual defection | result summary | float | 0 | 1 |
| $p_n$ | The probability of a player's action being flip at each interaction | trial summary | float | 0 | 1 |
| $n$ | The number of turns | trial summary | integer | 1 | 200 |
| $p_e$ | The probability of a match ending in the next turn | trial summary | float | 0 | 1 |
| $N$ | The number of strategies in the tournament | trial summary | integer | 3 | 195 |
| $k$ | The number of repetitions of a given tournament | trial summary | integer | 10 | 100 |

## The effect of strategy features on performance

For each strategy, we have a variety of features as described in Table 3. These features capture measures related to a strategy's behavior in the tournaments it competed in, as well as intrinsic properties, such as whether a strategy is deterministic or stochastic. The correlation coefficients between the features for performance evaluation, the median score and the median normalised rank are given by Table 4. The correlation coefficients between all features have also been calculated and a graphical representation can be found in the Supplementary Material (S1 Text).

In standard tournaments, the features $CC$ to $C$, $C_r$, $C_r/C_{max}$, and the cooperating ratio compared to $C_{median}$ and $C_{mean}$ have a moderately negative effect on the normalized rank (a smaller rank is better) and a moderate positive effect on the median score. The SSE error and the $DD$

**Table 4. Correlations between the features of Table 3 and the normalised rank and the median score.** For each type of tournament, standard, noisy, probabilistic ending, and noisy probabilistic ending, we conduct a correlation analysis. For each tournament, we check the correlation between each feature used in our analysis and the normalized random and median scores. Note that the correlation coefficients are calculated using Spearman's rank correlation coefficient. A negative value indicates a negative correlation, and in the case of the normalized rank, a smaller rank translates to a better position in the tournament.

| | Standard | | Noisy $p_n \leq 0.1$ | | Probabilistic ending $p_e \leq 0.1$ | | Noisy probabilistic ending | |
| --- | --- | --- | --- | --- | --- | --- | --- | --- |
| | r | median score | r | median score | r | median score | r | median score |
| $CC$ to $C$ rate | -0.501 | 0.501 | -0.210 | 0.194 | -0.336 | 0.348 | 0.087 | 0.015 |
| $CD$ to $C$ rate | 0.226 | -0.199 | 0.337 | -0.235 | 0.458 | -0.352 | 0.609 | -0.372 |
| $DC$ to $C$ rate | 0.127 | -0.100 | 0.227 | -0.111 | 0.164 | -0.105 | 0.410 | -0.203 |
| $DD$ to $C$ rate | 0.412 | -0.396 | 0.549 | -0.391 | 0.433 | -0.378 | 0.615 | -0.407 |
| $C_r$ | -0.323 | 0.383 | 0.298 | -0.051 | -0.060 | 0.160 | 0.595 | -0.213 |
| $C_{max}$ | 0.000 | 0.050 | -0.000 | 0.244 | -0.000 | 0.079 | -0.000 | 0.296 |
| $C_{min}$ | 0.000 | 0.085 | 0.000 | -0.070 | 0.000 | 0.128 | 0.000 | 0.000 |
| $C_{median}$ | 0.000 | 0.209 | 0.000 | 0.572 | -0.000 | 0.324 | 0.000 | 0.667 |
| $C_{mean}$ | 0.000 | 0.229 | -0.000 | 0.583 | -0.000 | 0.354 | -0.000 | 0.689 |
| $C_r / C_{max}$ | -0.323 | 0.381 | 0.307 | -0.076 | -0.060 | 0.156 | 0.608 | -0.246 |
| $C_{min} / C_r$ | 0.109 | -0.080 | -0.141 | -0.011 | 0.024 | 0.029 | -0.335 | 0.092 |
| $C_r / C_{median}$ | -0.330 | 0.353 | 0.326 | -0.258 | -0.065 | 0.111 | 0.614 | -0.464 |
| $C_r / C_{mean}$ | -0.331 | 0.357 | 0.325 | -0.228 | -0.066 | 0.114 | 0.617 | -0.431 |
| $N$ | -0.000 | -0.009 | -0.000 | -0.017 | -0.000 | 0.011 | 0.000 | 0.139 |
| $k$ | -0.000 | -0.002 | -0.000 | -0.003 | -0.000 | 0.010 | -0.000 | 0.035 |
| $n$ | -0.000 | -0.125 | -0.000 | -0.392 | - | - | - | - |
| $p_n$ | - | - | 0.000 | -0.244 | - | - | 0.000 | -0.272 |
| $p_e$ | - | - | - | - | 0.000 | 0.257 | 0.000 | 0.568 |
| Make use of game | -0.003 | -0.022 | -0.047 | 0.014 | -0.046 | 0.022 | -0.110 | 0.057 |
| Make use of length | -0.158 | 0.124 | -0.224 | 0.139 | -0.173 | 0.128 | -0.206 | 0.115 |
| SSE | 0.473 | -0.452 | 0.589 | -0.412 | 0.458 | -0.418 | 0.571 | -0.383 |
| stochastic | 0.006 | -0.024 | 0.010 | -0.007 | -0.001 | 0.001 | -0.001 | 0.002 |
| memory usage | -0.098 | 0.108 | -0.080 | 0.114 | - | - | - | - |

to $C$ rate have the opposite effects. Thus, in standard tournaments, behaving cooperatively corresponds to a more successful performance. Even though being nice generally pays off, that does not hold against defective strategies. Being more cooperative after a mutual defection, that is not retaliating, is associated with lesser overall success in terms of normalized rank. Compared to standard tournaments, in both noisy and noisy probabilistic ending tournaments, the higher the rates of cooperation, the lower a strategy's success and median score. A strategy would not want to cooperate more than both the mean and median cooperator in such settings. In probabilistic ending tournaments, the cooperation rate of the winners and its relative comparison to the cooperation rates of the tournament have no effect. The only features that have an effect are the $CD$ to $C$ rate, which is the tendency of a strategy to forgive, and the SSE rate, which has a positive effect on the normalized rank.

A multivariate linear regression has been fitted to model the relationship between the features and the normalized rank. Based on the graphical representation of the correlation matrices given in the Supplementary Material (S1 Text), several features are highly correlated and have been removed before fitting the linear regression model. The features included are given in Table 5 alongside their corresponding $p$ values in distinct tournaments and their regression coefficients. The $CD$ to $C$ rate has a positively statistically significant effect on the normalized rank across all tournament types. This suggests that being generous tends to lower one's performance. In the case of probabilistic ending tournaments, the coefficient of the $CD$ to $C$ rate

**Table 5. Results of multivariate linear regressions with $r$ as the dependent variable.** $R$ squared is reported for each model. The $R$ scores of the fitted models indicate their capability to explain some of the variation in the median rank. Most of the features have a statistically significant effect on the normalized rank. A multivariate linear regression has also be fitted on the median score. The coefficients and $p$ values of the features can be found in Supplementary Material (S1 Text). Both approaches lead to similar conclusions.

| | Standard | | Noisy $p_n \leq 0.1$ | | Probabilistic ending $p_e \leq 0.1$ | | Noisy probabilistic ending | |
|---|---|---|---|---|---|---|---|---|
| | R adjusted: 0.541 | | R adjusted: 0.373 | | R adjusted: 0.457 | | R adjusted: 0.537 | |
| | Coefficient | p-value | Coefficient | p-value | Coefficient | p-value | Coefficient | p-value |
| constant | 0.695 | 0.000 | 0.560 | 0.000 | 0.627 | 0.000 | 0.345 | 0.005 |
| $CC$ to $C$ rate | -0.042 | 0.000 | -0.163 | 0.000 | -0.046 | 0.000 | 0.032 | 0.029 |
| $CD$ to $C$ rate | 0.297 | 0.000 | 0.064 | 0.000 | 0.412 | 0.000 | 0.292 | 0.000 |
| $DC$ to $C$ rate | 0.198 | 0.000 | 0.142 | 0.000 | 0.193 | 0.000 | 0.193 | 0.000 |
| SSE | 0.258 | 0.000 | 0.328 | 0.000 | 0.190 | 0.000 | 0.228 | 0.000 |
| $C_{max}$ | -0.068 | 0.000 | -0.048 | 0.214 | -0.040 | 0.347 | -0.011 | 0.936 |
| $C_{min}$ | -0.161 | 0.000 | -0.029 | 0.367 | -0.049 | 0.017 | 0.008 | 0.912 |
| $C_{mean}$ | 0.117 | 0.000 | -0.133 | 0.000 | -0.159 | 0.000 | -0.468 | 0.000 |
| $C_{min} / C_r$ | 0.057 | 0.000 | -0.006 | 0.322 | 0.054 | 0.000 | 0.034 | 0.099 |
| $C_r / C_{mean}$ | -0.468 | 0.000 | -0.073 | 0.000 | -0.150 | 0.000 | 0.094 | 0.000 |
| $k$ | 0.000 | 0.325 | 0.000 | 0.965 | 0.000 | 0.079 | 0.000 | 0.065 |
| $n$ | 0.000 | 0.000 | - | - | - | - | - | - |
| memory usage | -0.010 | 0.000 | -0.008 | 0.000 | - | - | - | - |
| $C_r / C_{median}$ | - | - | 0.069 | 0.001 | -0.142 | 0.000 | - | - |
| $p_n$ | - | - | -0.131 | 0.010 | - | - | -0.278 | 0.048 |
| $p_e$ | - | - | - | - | -0.071 | 0.016 | 0.320 | 0.024 |

is the highest, indicating that one should be more provocative in this setting. Similarly, the SEE error rate has a positive effect on the normalized rank, suggesting that being extortionate pays off, especially in noisy tournaments. The measures of cooperation, $C_r$ and $C_r/C_{max}$, also exhibit a significant effect. In noisy probabilistic ending tournaments, this effect is positive; however, the coefficient is very close to zero. In other tournament types, the effect is negative, indicating that one should aim to be less cooperative than the mean cooperator of the tournament. However, we cannot interpret the result as suggesting that a strategy should be as uncooperative as possible.

The results presented here suggest that generosity/provocation and a strategy's cooperation rate, particularly in comparison to the tournament averages, are significant features. The analysis suggests that strategies should be more generous in noisy tournaments and less generous in probabilistic ending tournaments. Moreover, strategies should aim to not cooperate more than the mean cooperator in their tournaments. We note the analysis is limited as we only consider a linear relationship between these parameters and the rank. To further investigate the effects of the parameters discussed in this section, we have conducted a more detailed analysis in the next section, focusing on the performances of the winners of the tournaments.

## Features of top performing strategies

In Fig 4, we present the distributions of the cooperation ratio and $C_r/C_{mean}$ for the winners of tournaments. A value of $C_r/C_{mean} = 1$ implies that the cooperation ratio of the winner was the same as the mean cooperating ratio of the tournament, and we observe that this occurs for most tournament types, apart from the case of noisy and probabilistically ending tournaments. In the case of probabilistic ending tournaments, there are several winners that cooperated much less than that, confirming the results of the previous section that defecting strategies can be winners in probabilistic ending tournaments. The distribution of the cooperation rates

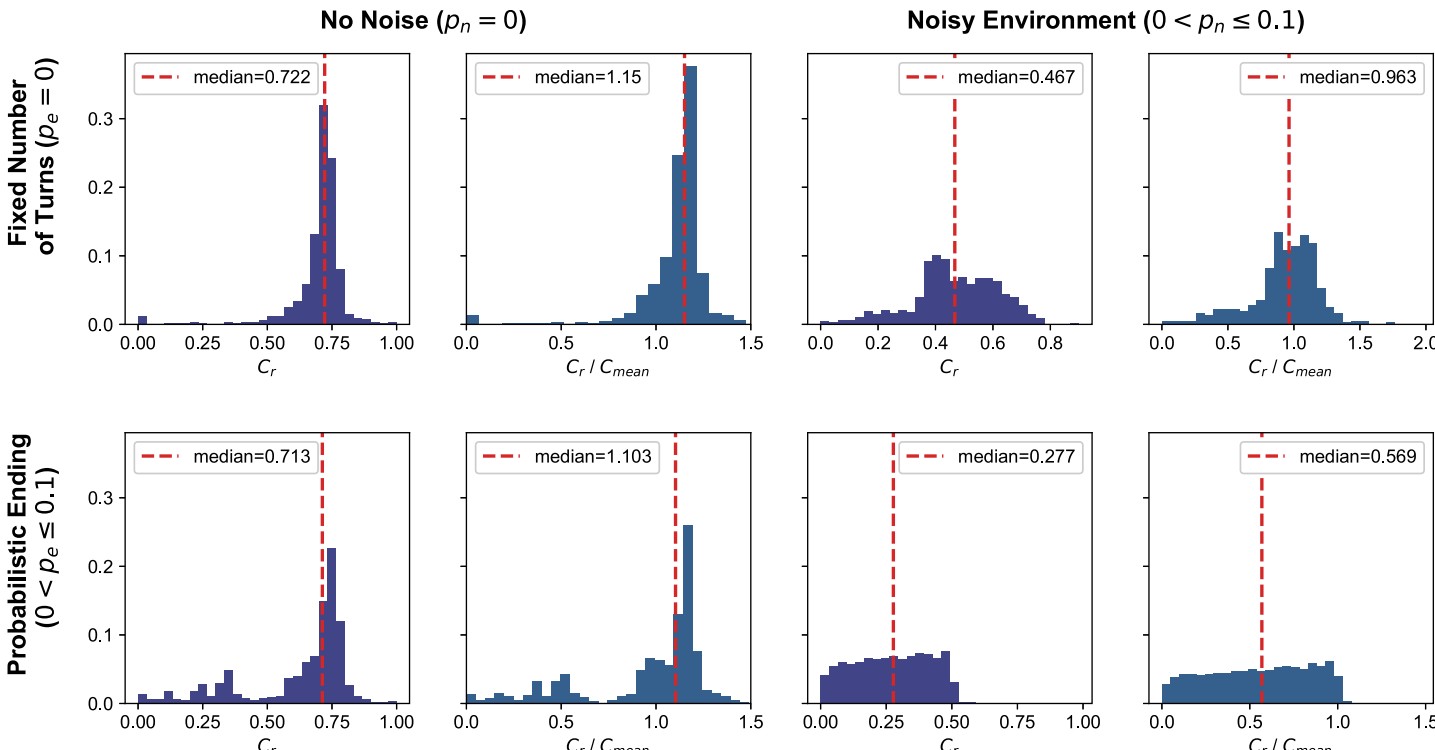

**Fig 4. Distributions of $C_r$ and $C_r/C_{mean}$ for the winners of tournaments.** In this distribution, we consider the winners of the tournaments, specifically the strategies that ranked first in each tournament. For each type of tournament, we plot the cooperation rate of the winner in the tournament they won, as well as the ratio of the winner's cooperation rate to that of the entire tournament. A value of $C_r/C_{mean} = 1$ implies that the cooperation ratio of the winner was the same as the mean cooperation ratio of the tournament.

showcases a high cooperation rate in standard tournaments and probabilistic ending tournaments. In tournaments with noise, we observe a much less cooperative behavior, which could result from strategies being cautious of potential flip actions by the co-player or strategies not suited for noise holding grudges against defections.

Analyzing the SSE distributions across different tournament types (Fig 5) suggests that successful strategies exhibit some extortionate behavior, though not consistently. ZDs are a set of strategies that are often envious, as they attempt to exploit their opponents. The winners of the tournaments considered in this work demonstrate envious behavior, but not to the extent observed in many ZDs. While the exact interactions between matches are not recorded here, the work of [23], which introduced the trained strategies appearing in the top-ranked strategies of Section, did record such interactions. In [23], it was shown that clever strategies managed to achieve mutual cooperation with stronger strategies while exploiting weaker ones. This could explain the clever winners in our analysis and the observed SSE distributions.

This might also be the reason why ZDs fail to appear in the top ranks—they attempt to exploit all opponents and cannot actively adapt back to mutual cooperation against stronger strategies, which requires a deeper memory. It's worth noting that ZDs tend to perform poorly in population games for a similar reason: they aim to exploit other players using ZDs, failing to form a cooperative subpopulation [36]. This makes them effective invaders but poor at resisting invasion.

Finally, we examine the distributions of the cooperation rates after the outcomes *CC*, *CD*, *DC*, and *DD*, as shown in Fig 6. In the case of cooperating after mutual cooperation, the results

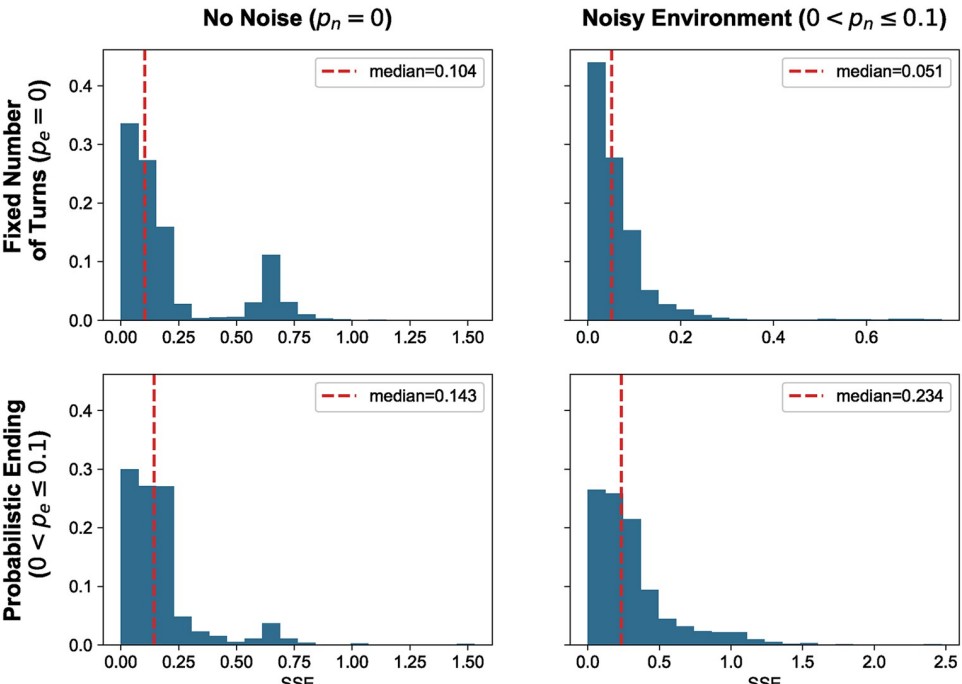

**Fig 5. Distributions of SSE error for the winners of tournaments.** Here, we again consider the winners of the tournaments, separated by type of tournament, and plot their SSE error. As a reminder, the SSE error indicates how closely a strategy behaves like a Zero-Determinant (ZD) strategy, and subsequently, in an extortionate way. An SSE value of 1 indicates no extortionate behavior at all, whereas a value of 0 indicates that a strategy is behaving as a ZD.

align with expectations; the distributions skew towards 1, indicating that the winners of the tournaments are more likely to cooperate after mutual cooperation. Regarding the *CD* outcome and the likelihood to cooperate after such a result, capturing generosity, the distributions skew towards 1/2, not 1, suggesting that strategies need to reduce their readiness to forgive. This aligns with the known result that Generous Tit For Tat generally outperforms `TFT` in most settings. In probabilistic ending tournaments, there is a peak at 0, suggesting that strategies should not be too generous in tournaments with short matches. Such a peak also appears in standard tournaments; however, not in tournaments with noise, where a strategy should be more generous.

Part of a strategy's envious behavior can be captured by the rate of *DC* to *C*. In noisy tournaments, winners are not too envious, but in tournaments without noise, we can see that winners behave in two ways. Some are a bit envious, whereas others are very envious. In the *DD* to *D*, we can observe that, expectedly, the results are skewed towards 0. However, there are winners that attempt to recover from a *DD* outcome. The remaining results are as expected, skewed towards 0.

## Discussion

This manuscript explores the performance of 195 strategies in the `IPD` in thousands of computer tournaments. The collection of computer tournaments presented here is the largest and most diverse in the literature. The 195 strategies are drawn from `Axelrod-Python` library and include strategies from the `IPD` literature. The computer tournaments encompass four different types. So, what is the best way to play the `IPD`? And is there a single dominant strategy for the `IPD`? There was not a single strategy within the collection of 195 strategies that

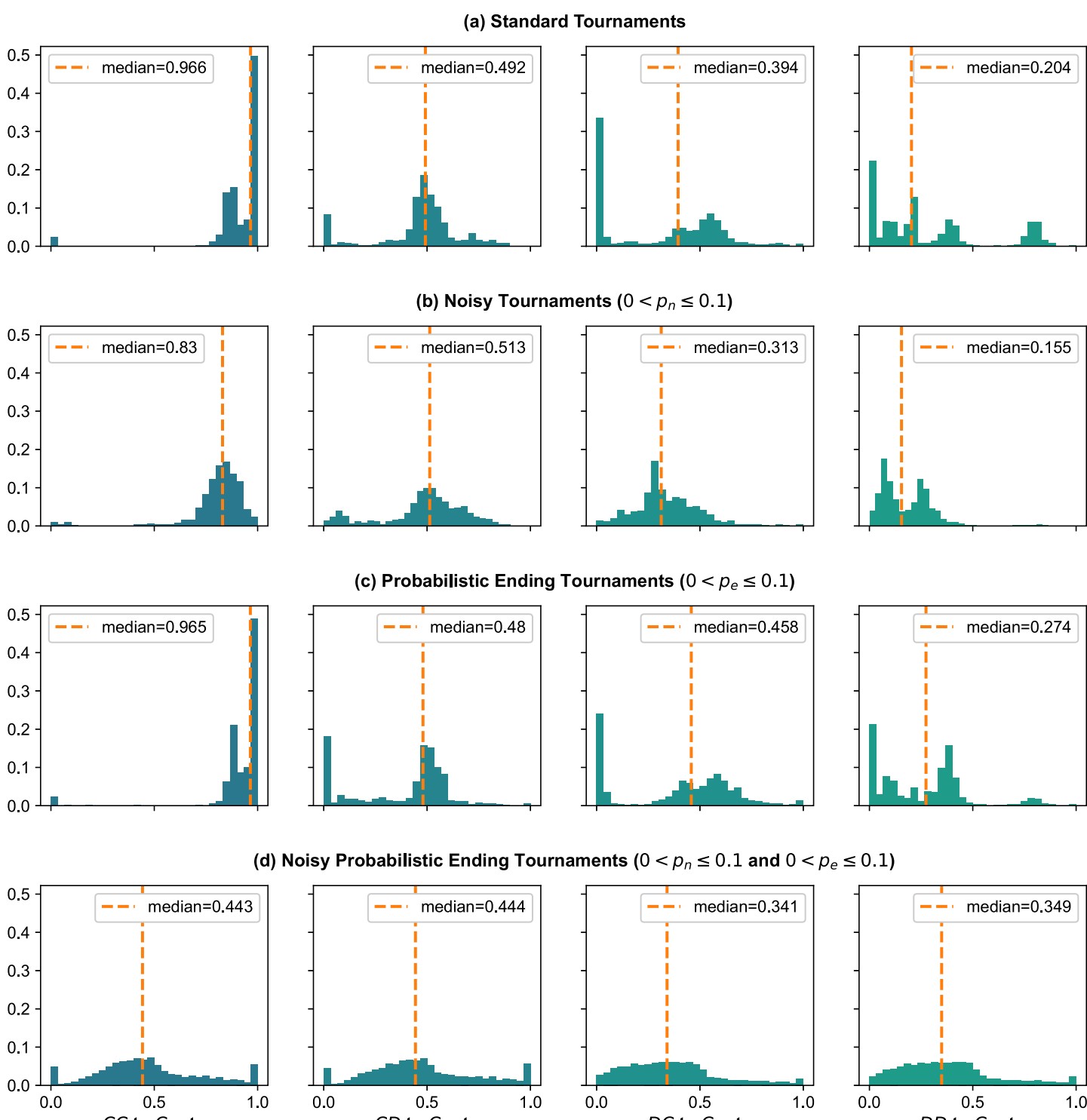

**Fig 6. Distributions of rates *CC* to *C*, *CD* to *C*, *DC* to *C*, and *DD* to *C* for the winners of tournaments.** The result summary from the tournaments records how often each strategy cooperated after each possible outcome of the previous round. Specifically, we analyze the probability with which a strategy chose *C* following the outcomes. Here, we plot the distributions of these probabilities for the winners of the tournaments. We separate them by type of tournament, from top to bottom: standard, noisy, probabilistic ending, and noisy probabilistic ending. From left to right, we plot the distributions of cooperation after the outcomes *CC*, *CD*, *DC*, and *DD*.

managed to perform well in all the tournament variations it competed in. A strategy ranking highly in a specific environment did not guarantee its success over different tournament types, with a few exceptions—strategies that generalize better. Already well-known in the AI/ML literature, adding noise to training data leads to more robust models [37]. We see that clearly here, where the strategies trained for noise (or designed for noise) tend to be better generalists. There were instances where a few strategies trained in narrow conditions outperformed more generalist strategies, as they tend to overfit. However, the strategies trained with noise perform well in general, whilst the strategies trained specifically on no noise or small subpopulations do not.

We also examined the best-performing strategies across various tournament types and analyzed their salient features. This demonstrated that there are properties associated with the success of strategies that contradict the originally suggested properties of Axelrod [1]. We showed that complex or **clever** strategies can be effective, whether trained against a corpus of possible opponents or purposely designed to mitigate the impact of noise such as the DBS strategy. Moreover, we found some strategies designed or trained for noisy environments were also highly ranked in noise-free tournaments which reinforces the idea that strategies' complexity/cleverness is not necessarily a liability, rather it can confer adaptability to a more diverse set of environments. We also showed that while the type of exploitation attempted by ZDs is not typically effective in standard tournaments, **envious** strategies capable of both exploiting and not their opponents can be highly successful. Based on the results of [23] this could be because they are selectively exploiting weaker opponents while mutually cooperating with stronger opponents. Highly noisy or tournaments with short matches also favoured envious strategies. These environments mitigated the value of being nice. Uncertainty enables exploitation, reducing the ability of maintaining or enforcing mutual cooperation, while triggering grudging strategies to switch from typically cooperating to typically defecting.

The features analysis of the best performing strategies demonstrated that a strategy should reciprocate, as suggested by Axelrod, but it should relax its readiness to do so and be more **generous**. For noisy environments this is inline with the results of [16–19], however, we also showed that generosity pays off even in standard settings, and that in fact the only setting a strategy would want to be too provocable is when the matches are not long. Forgiveness as defined by Axerlod was not explored here. This was mainly because the two round states were not recorded during the data collection. This could be a topic of future work that examines the impact of considering more rounds of history. The features analysis also concluded that there is a significant importance in **adapting to the environment**, and more specifically, to the mean cooperator. In most tournament types, the winner of the tournament was also the average cooperator. Even in tournaments with short matches where defecting behavior could secure a win, a large number of winners were average cooperators.

This could potentially explain the early success of TFT. TFT naturally achieves a cooperation rate near $C_{\mathrm{mean}}$ by virtue of copying its opponent's last move while also minimizing instances where it is exploited by an opponent (cooperating while the opponent defects), at least in non-noisy tournaments. It could also explain why Tit For $N$ Tats does not fare well for $N > 1$—it fails to achieve the proper cooperation ratio by tolerating too many defections.

Our results may also help explain the historically unexpected effectiveness of memory-one strategies [38]. The success of these strategies contradicts the intuitive assumption that a longer memory and therefore more information would yield better strategic performance [39]. Given that among the important features associated with success are the relative cooperation rate to the population average and the four memory-one probabilities of cooperating conditional on the previous round of play, these features can be optimized by a memory-one strategy such as TFT. Usage of more history becomes valuable when there are exploitable opponent patterns.

This is indicated by the importance of SSE as a feature, showing that the first-approximation provided by a memory-one strategy is no longer sufficient. These results highlight a central idea in evolutionary game theory in this context: the fitness landscape is a function of the population (where fitness in this case is tournament performance) [40]. While that may seem obvious now, it shows why historical tournament results on small or arbitrary populations of strategies have so often failed to produce generalizable results.

To this end, many strategies, such as Win-Stay-Lose-Shift and Generous Tit For Tat, emerged due to their strong performance in evolutionary dynamics. Axelrod's original work relied on computer tournaments, so we chose to remain consistent with this approach, as a comprehensive study like ours had not yet been undertaken. However, evolutionary settings would be an exciting direction for future study.

Overall, the five properties successful strategies need to have in a `IPD` competition based on the analysis that has been presented in this manuscript are:

1.  Be "nice" in non-noisy environments or when game lengths are longer

2.  Be provocable in tournaments with short matches, and generous in tournaments with noise

3.  Be a little bit envious

4.  Be clever

5.  Adapt to the environment (including the population of strategies).

The results presented here were based only on a subset of the whole data we have collected. The analysis of the full dataset is discussed in the Supplementary Material (S1 Text). However, we can see that the general results of our work remain the same. In the Supplementary Material (S1 Text), we also evaluate the importance of features using a random forest classifier and a clustering approach. The results of these analyses are also in line with the results presented here.

The data set described in this work contains the largest number of `IPD` tournaments, to the authors knowledge. The raw data set is available at [41] and the processed data at [42]. Further data mining could be applied and provide new insights in the field.

## Supporting information

**S1 Text. Supplementary material.** This document provides details of further analysis, a summary of all parameters and notations used in the manuscript, and a comprehensive list of all strategies considered in this work.
(PDF)

## Acknowledgments

A variety of open-source software have been used in this work. The authors would like to express their gratitude to the open-source software community, whose invaluable contributions significantly enhanced the development and execution of this research. Namely, the authors would like to thank the developers of the following software packages: Axelrod-Python library for IPD simulations, the Matplotlib library for visualisation, The Numpy library for data manipulation, and finally the scikit-learn library for data analysis.

## Author Contributions

**Conceptualization:** Nikoleta E. Glynatsi, Vincent Knight, Marc Harper.

**Formal analysis:** Nikoleta E. Glynatsi.

**Investigation:** Nikoleta E. Glynatsi.

**Methodology:** Nikoleta E. Glynatsi, Vincent Knight, Marc Harper.

**Software:** Nikoleta E. Glynatsi.

**Visualization:** Nikoleta E. Glynatsi.

**Writing – original draft:** Nikoleta E. Glynatsi, Vincent Knight, Marc Harper.

**Writing – review & editing:** Nikoleta E. Glynatsi, Vincent Knight, Marc Harper.

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
