## [Decision Letter · Decision Letter 0]

8 Oct 2024

Dear Dr. Glynatsi,

Thank you very much for submitting your manuscript "Properties of Winning Iterated Prisoner's Dilemma Strategies" for consideration at PLOS Computational Biology. As with all papers reviewed by the journal, your manuscript was reviewed by members of the editorial board and by several independent reviewers. The reviewers appreciated the attention to an important topic. Based on the reviews, we are likely to accept this manuscript for publication, providing that you modify the manuscript according to the review recommendations.

Sincerely,

Alexandre V. Morozov, Ph.D.

Academic Editor

PLOS Computational Biology

Tobias Bollenbach

Section Editor

PLOS Computational Biology

Reviewer's Responses to Questions

**Comments to the Authors:**

Reviewer #1: Title: Properties of Winning Iterated Prisoner's Dilemma Strategies.

Summary: This paper explores the performance of strategies in the IPD. This research analyses 195 strategies across thousands of computer tournaments, examining what makes certain strategies more successful in diverse IPD environments. The study used four different types of tournaments: Standard tournaments, Noisy tournaments, Probabilistic ending tournaments and Noisy probabilistic ending tournaments. They find that no single strategy excels in all settings. However, several key traits appear in successful strategies, particularly the ability to adjust to the environment and population dynamics.

Thanks for this submission, I find the text easy to read and all the concepts well explained. I still feel there are a few bits that increase its robustness:

1. This seems like a low-hanging fruit idea for someone like Axelrod-Python. If they have all of these strategies, what is the best, after all, they maintain and add new strategies to the library, why do you think this hasn't been done before? I feel the data analysis and the interpretation is sound, but maybe can you add other efforts to do the same, and what do you do differently?

2. You mention somewhere that there are evolutionary approaches to this, specifically evolutionary game theory approaches. Maybe adding why you chose to do computer simulations instead? (I know that EGT has some constraints, but highlight why you do how you do it to overcome those constraints).

3. I don't know if it's enough to say that you used Axelrod-Python, but maybe also add how these tournaments are performed? I haven't used the tool, so I don't know how they simulate those exactly. I feel the library itself is central to how the results are generated, so maybe expand a bit on that.

4. Check for references when you make statements. For example, you say:

These results highlight a central idea in evolutionary game theory in this context: the fitness landscape is a function of the population (where fitness in this case is tournament performance) (REF)

Similarly, our results could suggest an explanation regarding the intuitively unexpected effectiveness of memory-one strategies historically. (Why is it unexpected? REF)

Those are two examples, but check for that kind of sentences.

5. I believe it's a matter of style, but I prefer if figures are self-explanatory. As you did with Figure 1, where you guide the reader through what is shown, Table 4 and Figures 3,4 and 5 lack some context in the figure labels, especially in figures like 5, some guidance is appreciated.

Overall, a very clear read, perhaps some context on the state of the art of these analyses is needed, but in general, a good work.

Reviewer #2: The authors uncover properties of winning strategies in the iterated Prisoner’s Dilemma (IPD), one of the most renowned paradigms for studying cooperation over the past decades. They consider a large collection of 195 strategies in thousands of computer tournaments and conduct a thorough analysis of their performance. Their conclusions refine the properties described by Axelrod: a successful strategy needs to be nice, provocable, generous, somewhat envious, clever, and adaptable to the environment.

There are several aspects of this work that I appreciate greatly. First, the inclusiveness of the study design is commendable: it not only includes all kinds of strategies from IPD literature but also incorporates variations in the tournaments (such as random noise and match length). Second, their findings challenge some of Axelrod’s suggestions, particularly the advice to “not be clever” and “not be envious.” Additionally, I find the property of being adaptable to the environment—explained by the authors as a strategy’s cooperation rate compared to the average cooperation rate in a tournament—very inspiring.

I have only three minor suggestions/concerns for the authors to consider:

1. On page 8, I suggest including the explanations for C_max, C_median, C_mean, and SSE error in the main text, as they currently seem to be defined only in figures and tables.

2. In Table 3 on page 9, there is a contradiction between “A strategy’s cooperating rate divided by the minimum” and “C_min/C_r.”

3. On page 13, the sentence “envious strategies capable of both exploiting and not their opponents…” appears to be incomplete.

I will be very happy to see this work accepted after the authors have addressed the points mentioned above.

**Have the authors made all data and (if applicable) computational code underlying the findings in their manuscript fully available?**

Reviewer #1: Yes

Reviewer #2: Yes

PLOS authors have the option to publish the peer review history of their article (what does this mean?). If published, this will include your full peer review and any attached files.

Reviewer #1: No

Reviewer #2: **Yes: **Xingru Chen

Figure Files:

Data Requirements:

Reproducibility:

References:

---

## [Editor Report · Decision Letter 1]

17 Nov 2024

Dear Dr. Glynatsi,

We are pleased to inform you that your manuscript 'Properties of Winning Iterated Prisoner's Dilemma Strategies' has been provisionally accepted for publication in PLOS Computational Biology.

Best regards,

Alexandre V. Morozov, Ph.D.

Academic Editor

PLOS Computational Biology

Tobias Bollenbach

Section Editor

PLOS Computational Biology

Feilim Mac Gabhann

Editor-in-Chief

PLOS Computational Biology

Jason Papin

Editor-in-Chief

PLOS Computational Biology

---

## [Editor Report · Acceptance letter]

4 Dec 2024

PCOMPBIOL-D-24-01356R1 

Properties of Winning Iterated Prisoner's Dilemma Strategies

Dear Dr Glynatsi,

I am pleased to inform you that your manuscript has been formally accepted for publication in PLOS Computational Biology. Your manuscript is now with our production department and you will be notified of the publication date in due course.

With kind regards,

Anita Estes
